Brief Communication

# Effects and avoidance of photoconversion-induced artifacts in confocal and STED microscopy

Anindita Dasgupta [1,2], Agnes Koerfer[1,2], Boštjan Kokot[3], Iztok Urbančič [3], Christian Eggeling [1,2,4] ✉ & Pablo Carravilla [2,5] ✉

Fluorescence microscopy is limited by photoconversion due to continuous illumination, which results in not only photobleaching but also conversion of fluorescent molecules into species of different spectral properties through photoblueing. Here, we determined different fluorescence parameters of photoconverted products for various fluorophores under standard confocal and stimulated emission depletion (STED) microscopy conditions. We observed changes in both fluorescence spectra and lifetimes that can cause artifacts in quantitative measurements, which can be avoided by using exchangeable dyes.

The performance of fluorescence microscopy is limited by photobleaching of fluorophores due to an enhanced reactivity from their excited states, resulting in decreased fluorescence (Fig. 1a,b). Photobleaching varies with experimental conditions and strongly increases with excitation light intensity[1,2]. Consequently, light-intense microscopy techniques such as confocal or super-resolution stimulated emission depletion (STED) microscopy are prone to photobleaching[3]. Illumination may also induce photoconversion into species of different fluorescence emission properties, such as shifted wavelength ranges, lifetime or quantum yield (and thus brightness). Specifically, the generation of species with blue-shifted emission, so-called photoblueing, has been reported[4–9]. Quantitative measurements, especially when relying on such fluorescence emission readouts, will suffer from photoconversion and thus further studies are required, especially on biasing effects and possible minimization protocols.

In this work, we measured the photophysical properties of photoconversion products directly at the microscope under typical imaging conditions. We evaluated the photoconversion susceptibility of various commonly used organic dyes. By using a combination of quantitative and spectroscopic microscopy techniques, we precisely determined changes of their emission spectra, lifetimes and brightness values after repeated illumination, and identified potential artifact sources and avoidance strategies in quantitative microscopy.

First, we performed measurements on the red-emitting organic dyes ATTO 647N, Abberior STAR RED and ATTO 655, commonly used in confocal and STED microscopy (Extended Data Fig. 1a). We quantified photoconversion-induced shifts of the emission spectrum using a customized confocal STED microscope equipped with a grating element and an array of photomultipliers, allowing us to record the emission spectrum. First, antibody-conjugated dyes were immobilized on a coverslip in phosphate-buffered saline (PBS) and illuminated with 640 nm laser light. We observed photobleaching of the original red fluorescence signal already within the first ten frames of illumination for all dyes (Fig. 1b). Photobleaching was correlated with the appearance of a lower wavelength emission signal (photoblueing, visualized by 561 nm excitation in a more orange detection window, Fig. 1b), yet only for Abberior STAR RED and ATTO 647N but not ATTO 655 (Fig. 1c–e and Extended Data Fig. 1b–d). Photoblueing was observed for different dye conjugations (Extended Data Fig. 1b), and less so for the free dyes (Extended Data Fig. 1c,d). Spectral unmixing at different illumination times revealed the emission spectra of the photoblued species, highlighting a 20–25 nm shift of the maxima for both dyes (Extended Data Fig. 1e–g), which made up as much as 50% of the emission on prolonged illumination (Extended Data Fig. 1h). These spectral shifts were smaller than those observed for other dyes[9].

The most important question is whether photoconversion products affect measurements, specifically whether they might dominate

[1]Institute for Applied Optics and Biophysics, Friedrich Schiller University Jena, Jena, Germany. [2]Leibniz Institute of Photonic Technology e.V., member of the Leibniz Centre for Photonics in Infection Research (LPI), Jena, Germany. [3]Jožef Stefan Institute, Ljubljana, Slovenia. [4]Jena Center for Soft Matter (JCSM), Jena, Germany. [5]Science for Life Laboratory, Department of Women's and Children's Health, Karolinska Institute, Solna, Sweden. ✉e-mail: christian.eggeling@uni-jena.de; pablo.carravilla@ki.se

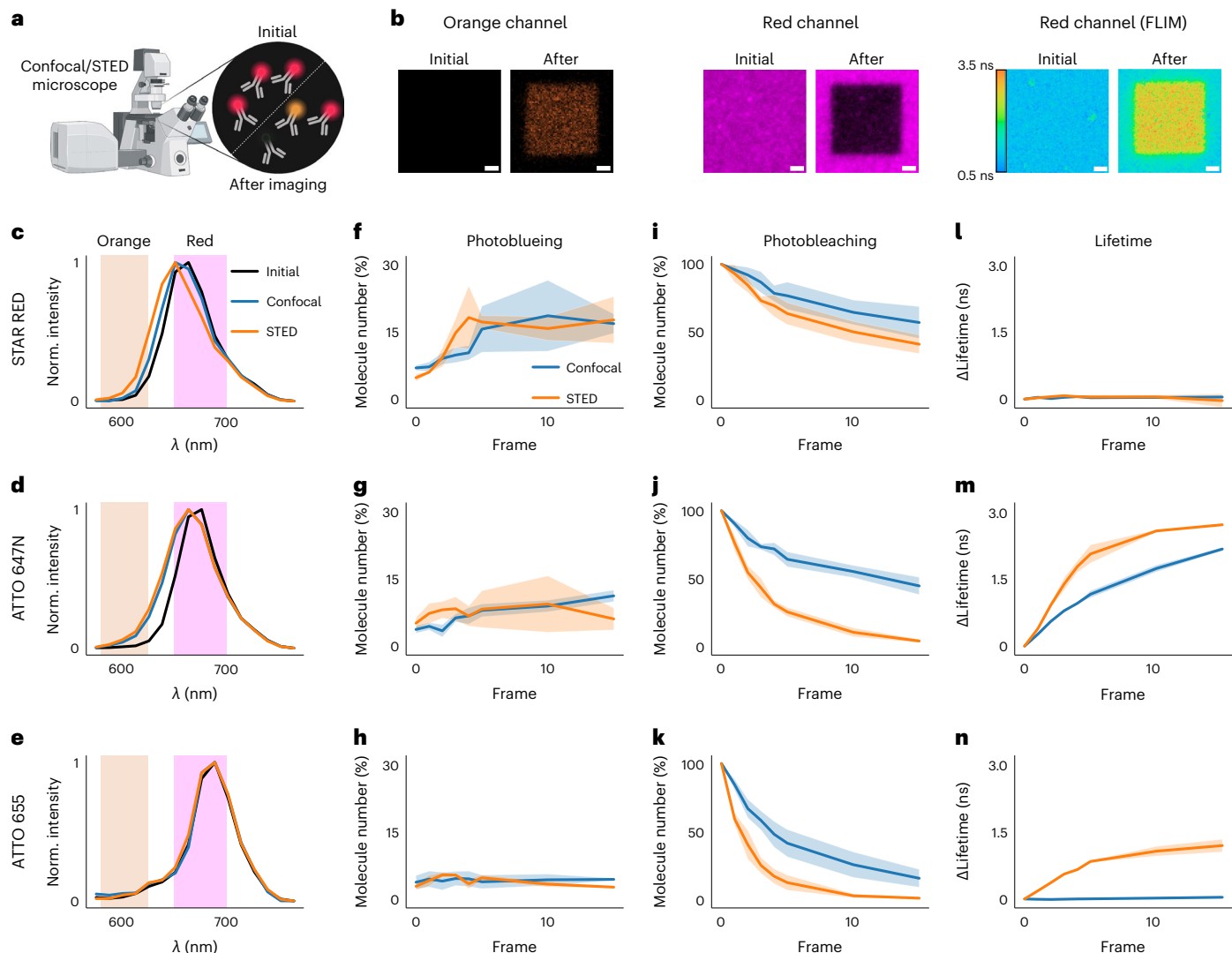

**Fig. 1 | Quantifying photoconversion products directly at the microscope.**
**a**, Illumination induces photobleaching (black in inset) and photoconversion (orange) of fluorophores (red). Panel created with BioRender.com.
**b**, Continuous illumination of a region of interest (5 × 5 µm²) of immobilized antibody-conjugated fluorophores in PBS using 640 nm excitation on a confocal microscope highlights photoblueing (left, fluorescence emission on 561 nm excitation) and photobleaching (middle, fluorescence emission on 640 nm excitation) for Abberior STAR RED, and changes in fluorescence lifetime for ATTO 647N (right, fluorescence lifetime on 640 nm excitation). Scale bars, 1 µm.
**c–e**, Emission spectrum of antibody-conjugated Abberior STAR RED (**c**), ATTO 647N (**d**) and ATTO 655 (**e**), before (Initial) and after ten frames of confocal or STED microscopy imaging. Spectra were recorded with 561 nm excitation.
**f–k**, FCS quantification of the average number of molecules N (relative to time point 0) in the microscope observation volume in the green or orange (488 nm excitation, **f–h**) and red (640 nm excitation, **i–k**) detection windows (relative percentage value compared to initial molecule number in the red channel) after the indicated number of confocal or STED microscopy imaging frames with 640 nm excitation as measured for lipid-conjugated Abberior STAR RED (**f,i**), ATTO 647N (**g,j**) and ATTO 655 (**h,k**) on an entirely illuminated SLB patch.
**l–n**, Changes of the average fluorescence lifetime as measured by FLIM on confocal or STED microscopy imaging of antibody-conjugated Abberior STAR RED (**l**), ATTO 647N (**m**) and ATTO 655 (**n**). Excitation power at the sample plane was 10 µW (640 nm) in all cases. STED (775 nm) laser power was 200 mW in **b–d** and 100 mW in **f–n**. Mean and standard deviation of three independent experiments are shown. Norm., normalized.

the signal in blue-shifted detection channels (Fig. 1b) that may originally be dedicated to other fluorophores in multicolor experiments. A key parameter here is the brightness of individual photoblued dyes. To precisely determine this parameter, we turned to fluorescence correlation spectroscopy (FCS), which reports on different molecular parameters such as single-molecule fluorescence brightness ($Q$) and the average number ($N$) of fluorescent molecules in the observation volume. We prepared µm-sized supported lipid bilayer (SLB) patches labeled with a fluorescent lipid analog tagged with ATTO 647N, Abberior STAR RED or ATTO 655. Continuous illumination with 640 nm laser light of a whole patch led to photoblueing of the embedded fluorophores, and we recorded FCS data of the

original and photoblued species (with 640 or 488 nm excitation, respectively) at different time points of the photoblueing process (Extended Data Fig. 2a–d), which allowed us to probe $Q$ and $N$ for both species. Values of $N$ revealed photoblueing of as many as 20% of the original molecules already after just ten illumination frames under confocal and STED conditions for both Abberior STAR RED and ATTO 647N, and again only negligible photoblueing for ATTO 655 (Fig. 1f–k and Extended Data Fig. 2e). However, the molecular brightness $Q$ of the photoconverted dyes on 488 nm excitation was 20–100 times lower than of the original and commonly used green and orange dyes (Extended Data Fig. 2f,g), highlighting the negligible contribution to the green FCS detection channel.

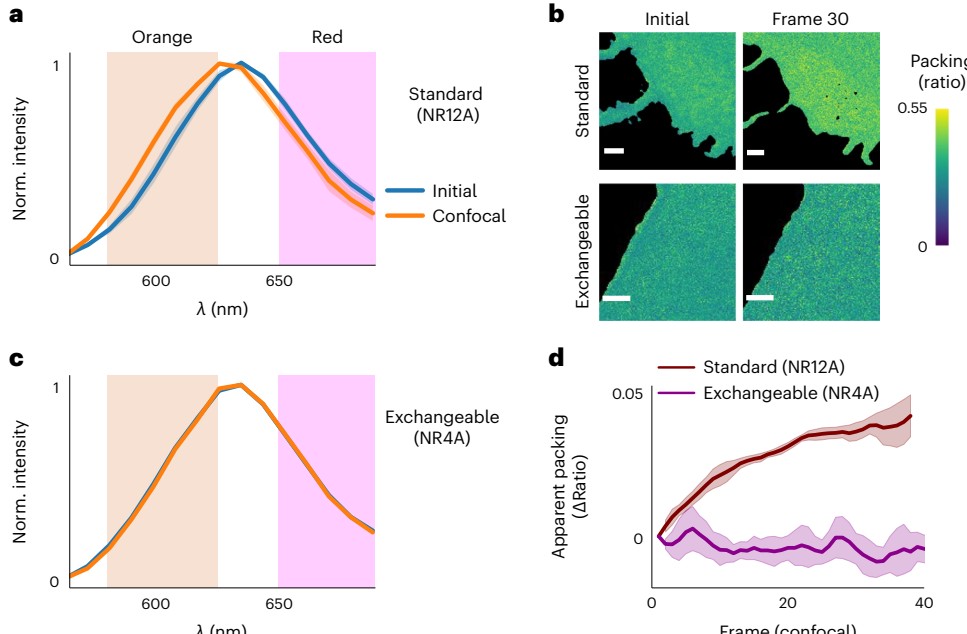

**Fig. 2 | Photoconversion induces artifacts in quantitative imaging, which is avoided by the use of exchangeable dyes. a**, Emission spectrum of the membrane lipid packing probe NR12A before and after 30 confocal imaging frames (561 nm excitation) in POPC SLBs. Mean and s.d. of six measurements in three independent experiments is shown. **b**, Ratiometric GP images of live cells shows a change in the measured lipid packing when using standard dyes (NR12A), which is not detected when using an exchangeable fluorophore (NR4A, PAINT mode). On the right are lipid packing images after 30 frames for the standard and exchangeable fluorophore, respectively. **c**, Emission spectra of the exchangeable probe NR4A, conditions as in **a**. **d**, Biased increase in apparent lipid packing of the plasma membrane of Ptk2 cells is measured when using the standard probe NR12A due to photoblueing, which is avoided with the exchangeable fluorophore NR4A. Shown is the relative change of GP value on confocal excitation. Mean and s.d. of three independent measurements are shown. Excitation power was 5 µW (561 nm) at the sample plane in all cases. Scale bars, 1 µm.

To explore possible mechanisms of the photoblueing pathway, we investigated dependencies on oxygen. Oxygen is known to have two main effects on the dyes' photophysics: (1) quenching of the triplet state and therefore fewer photoreactions out of that state, and (2) increased general photoreactivity due to interaction with the solved oxygen, especially when in its singlet state[2,10,11]. We consistently observed that photoblueing was suppressed as a photoreaction in the absence of oxygen (Extended Data Fig. 2h,i), as highlighted before for other dyes[6,9]. On one hand, this indicates that the photoblueing pathway involves photooxidation (such as in photooxidative N-dealkylation processes[11] or through singlet oxygen produced by quenching of the dyes' triplet state[10]) and barely evolves directly through the triplet state, and on the other hand, it suggests that photoblueing is only one of many possible photoreaction pathways. Our FCS data highlight a correlation between the extent of photoblueing and the amount of intersystem crossing and thus triplet yield, being lowest for ATTO 655 and largest for Abberior STAR RED (Extended Data Fig. 2d). Since singlet oxygen is generated through triplet–triplet energy transfer, we speculate that photoblueing is fostered via this mechanism.

Another important parameter is the fluorescence lifetime of a fluorophore. Fluorescence lifetime imaging microscopy (FLIM) is a popular tool to measure molecular interactions in cells through Förster resonance energy transfer. To assess the potential lifetime changes induced by excitation and STED laser illumination, we performed FLIM measurements of the antibody-conjugated dyes. While we observed no lifetime changes for Abberior STAR RED (Fig. 1l), we revealed a >1 ns increase in lifetime after just five imaging frames in the case of ATTO 647N (Fig. 1m and Extended Data Fig. 3a,b), and under STED conditions for ATTO 655 (Fig. 1n). These data were recorded on antibody-conjugated dyes. Here, a known phenomenon arises from high labeling degrees (more than one dye conjugated to a single antibody): quenching effects between the dyes in close proximity, and thus fluorescence lifetime changes[12,13]. Illumination will in this case result in an initial partial bleaching of one of the dyes and thus a rise in lifetime and an initial rise in fluorescence intensity followed by the drop due to photobleaching of all dyes, as observed in our case (Extended Data Fig. 3d,e). This fact and the observation that the lifetime change was only partially inhibited on enzymatic oxygen depletion (Extended Data Fig. 3c), highlighted that lifetime changes are not a consequence of photoblueing but rather due to the outlined partial photobleaching. Consistently, we did not observe a lifetime increase on illumination for lipids conjugated to single dyes, which instead showed a minor lifetime decrease (Extended Data Fig. 3f,g). Consequently, single-dye conjugation will serve as an artifact mitigation strategy in FLIM experiments.

Finally, we turned to the effect of photoblueing in spectral experiments. Spectral changes are commonly monitored to quantify biophysical properties such as membrane lipid packing or tension[14,15]. An increase in membrane packing is monitored by a blue shift of the emission spectrum of a few nanometers, usually measured through ratiometric imaging[14,16]. Our experiments revealed that the environment-sensitive membrane probe NR12A (ref. 17) underwent notable photoblueing (Fig. 2a), which obviously falsely reports an increase in plasma membrane packing in live Ptk2 cells (Fig. 2b and Extended Data Fig. 4a,b). Indeed, packing values were restored by replenishing the probe, confirming that the observed blue shift in the emission spectrum was a photoconversion artifact, and not an effect on membrane packing itself due to phototoxicity (Extended Data Fig. 4c). Recently, it was shown that exchangeable dyes, which only temporarily bind to their target (as in PAINT microscopy[18,19]), can circumvent photobleaching-induced signal loss and allow for long-term microscopy measurements[20]. Indeed, NR4A, an exchangeable version of the biophysical probe NR12A (ref. 17), showed no spectral shift on irradiation with excitation and STED lasers (Fig. 2c), since photoconverted

dyes are readily substituted, and consequently it accurately reported on lipid packing over long measurements (Fig. 2b,d).

In conclusion, we demonstrated that photoreactions may induce changes of fluorescence parameters, such as emission spectrum and lifetime, and vary between dyes and experimental conditions. One photoreaction pathway is photoblueing, creating new blue-shifted dyes. These photoconverted dyes showed lower brightness and thus have a minor effect on single-molecule based multicolor ensemble measurements. Yet, care must be taken for microscopy approaches relying on the observation of isolated single molecules such as single particle tracking or localization[9]. On the other hand, changes in fluorescence lifetime were observed from partial photobleaching of molecules with multiple dye conjugations, as is the case for many fluorescently tagged antibodies. Oxygen dependencies of photoblueing and lifetime changes highlighted the existence of many different photoreaction pathways, where photooxidation and partial bleaching are only two of many possible processes. Caution has to be taken when a high number of photoconverted fluorophores promotes an overall shift in the fluorescence lifetime or blue shift, leading to a potential bias in purely ensemble-based quantitative microscopy experiments such as Förster resonance energy transfer, FLIM or environment sensing, which can be circumvented by the use of single-dye conjugation and exchangeable dyes, respectively.

## Online content

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

## Methods

### Dyes
Fluorescent dyes and their conjugated versions were acquired from the following manufacturers: Abberior STAR RED goat antimouse IgG (Abberior, STRED-1001), Abberior STAR RED DPPE (Abberior, STRED-0200), Abberior STAR RED Streptavidin (STRED-0120), ATTO 647N free acid (ATTO-TEC, AD 647N-21), Anti-Mouse-IgG—Atto 647N antibody produced in goat (Sigma-Aldrich, 50185), ATTO 647N-DPPE (ATTO-TEC, AD 647N-151), ATTO 655 free acid (ATTO-TEC, AD 655-21), Anti-Mouse IgG—Atto 655 antibody produced in goat (Sigma-Aldrich, 50283), ATTO 655-DPPE (ATTO-TEC, AD 655-151), MemGlow NR12A Membrane Polarity Probe (Cytoskeleton, Inc., MG07), MemGlow NR4A Membrane Polarity Probe (Cytoskeleton, Inc., MG06), Abberior STAR 600-DPPE (Abberior), TopFluor Cholesterol (Avanti Polar Lipids, 810255) and ATTO 488-DPPE (ATTO-TEC, AD 488-151).

### Sample preparation
**Immobilized dyes.** To immobilize dyes on glass surfaces for FLIM, round 25 mm no. 1.5 coverslips (VMR, catalog no. 631-0172P) were mounted on Attofluor Cell Chambers (Thermo Fisher Scientific, catalog no. A7816). For spectral imaging, dyes were immobilized on μ-Slide Eight-Well Glass-Bottom coverslips (Ibidi, catalog no. 80827). Dye concentrations for spectral imaging were: 500 nM ATTO 647N free acid, 900 nM ATTO 655 free acid, 8 μg ml$^{-1}$ Abberior STAR RED goat antimouse IgG, 10 μg ml$^{-1}$ Anti-Mouse-IgG—Atto 647N, 8 μg ml$^{-1}$ Anti-Mouse IgG—Atto 655. Concentrations for FLIM were: 2 μg ml$^{-1}$ Abberior STAR RED goat antimouse IgG, 1.6 μg ml$^{-1}$ Anti-Mouse-IgG—Atto 647N 0.8 mg ml$^{-1}$ and 8 μg ml$^{-1}$ Anti-Mouse IgG—Atto 655. Dye samples were diluted in PBS (pH 7.4), incubated for 10 min at room temperature in the imaging chamber, washed three times with PBS and subsequently imaged. For experiments performed in the absence of oxygen, after washing, PBS was exchanged with an enzymatic oxygen scavenging buffer[21] consisting of 10% (m/v) glucose, 500 μg ml$^{-1}$ glucose oxidase and 40 μg ml$^{-1}$ dissolved in 50 mM Tris-HCl, pH 8.0. The buffer was freshly prepared for every measurement.

**SLBs.** For FCS and FLIM measurements of lipid-conjugated dyes, SLBs were prepared in PBS to ensure dye mobility and resulting intensity fluctuations. To prepare SLBs, giant unilamellar vesicles (GUVs) were prepared using the electroformation method. Then 1-palmitoyl-2-oleoyl-glycero-3-phosphocholine (POPC) (Avanti Polar Lipids, catalog no. 850457P) lipid stock solutions were prepared at a final concentration of 1 g l$^{-1}$ in chloroform. The lipid solution (5 μl) was spread on two parallel platinum wires and these were dipped in a home-made polytetrafluoroethylene chamber filled with 400 μl of a 300 mM sucrose solution. GUVs were formed by connecting the wires to a function generator and applying a 2 V, 10 Hz alternating current for 1 h, followed by a 2 V, 2 Hz current for 30 min. GUVs were collected with a 1,000 μl plastic pipette tip; to avoid GUVs rupture, the tip diameter was widened by cutting it roughly 2 cm from the end. GUVs were transferred to a 2 ml tube and labeled with lipid dye analogs (100 nM for Abberior Starred-DPPE, 40 nM for ATTO 655-DPPE, 2 nM ATTO 647N-DPPE, 10 nM for Abberior STAR 600-DPPE and ATTO 488-DPPE, 20 nM Top-Fluor Cholesterol) for 5 min at room temperature. Then, 50 μl of GUVs were transferred to a plasma-treated (10 s) coverslip mounted in an Attofluor Cell Chamber prefilled with 950 μl of PBS. GUVs were left to sink for 10 min and SLB patches formed on contact of the GUVs with the plasma-treated surface. Last, three washing steps with 500 μl of PBS buffer were performed to discard unbound GUVs. Experiments in the absence of oxygen were performed as described in the 'Sample preparation: Immobilized dyes' section.

**Cells.** PtK2 potoroo kidney cells (American Type Culture Collection, catalog no. CCL-56) were grown at 37 °C and 5% CO$_2$ in DMEM (Carl Roth, catalog no. 9007.1) supplemented with 10% fetal bovine serum (Sigma-Aldrich, catalog no. F4135). Cells were subcultured at a 1/3 ratio every 3 days and kept for 30 passages. For imaging experiments, cells were seeded in μ-Slide Eight-Well Glass Bottom (Ibidi, catalog no. 80827) coverslips 24–48 h before imaging. The day of the experiment, cells were washed twice with Leibovitz's L-15 medium (Thermo Fisher Scientific, catalog no. 11415064), labeled with 20 nM NR12A or 400 nM NR4A, and subsequently imaged using L-15 as buffer.

### Spectral imaging
Spectral confocal and STED microscopy imaging was performed with a customized Abberior Instruments Expert Line laser scanning STED microscope built around an Olympus IX83 inverted body using an Olympus UplanSApo water immersion objective (×60/numerical aperture (NA) 1.2). 5 × 5 μm$^2$ regions of interest showing immobilized dye signal were imaged for the indicated number of frames using standard confocal and STED imaging conditions, namely 10 μW 640 nm/100 mW 775 nm STED laser power at the sample plane, 10 μs dwell time, 50 nm pixel size and repetition frequency of 40 MHz. To collect emission spectra, 488 and 561 nm laser lines were used with a 10 μW power as measured at the sample plane. The fluorescence was dispersed by a grating element onto 16 channels (12.5 nm spectral width per channel) of a GaAsP photomultiplier detector (MW FLIM, Becker & Hickl) and the signal sent to a time-correlated single-photon-counting device (catalog no. SPC-150, Becker & Hickl). All the data were acquired using iMSPECTOR v.6.3.15517 software (Abberior Instruments). The powers in the sample plane were determined by measuring the power at the objective back focal plane by passing light through a manual shutter closed to the size of the objective back aperture and then by focusing onto a Thorlabs Power Meter S120C. To extract the peak wavelengths and perform unmixing of spectral datasets, raw image files were converted to TIFF format and then fitted with a one- or two-component log-normal function using a custom Wolfram Mathematica script[22,23]. First, the background-subtracted spectral data of the dyes before irradiation were fitted with a one-component function to estimate the spectral parameters of the original dye, namely peak position ($\lambda_{max}$) and approximate full-width at half-maximum ($w$) (the asymmetry parameter ($a$) was fixed to 0.24). Then, to estimate the $\lambda_{max}$ and relative fraction of the photoblued species (component 2), the spectral data were fitted with a two-component function by fixing the parameters corresponding to the red dye (component 1).

### FCS
Measurements were performed in PBS. To deplete oxygen, an enzymatic scavenging buffer was used ('Sample preparation: Immobilized dyes' section). FCS data were acquired with an Abberior STEDYCON STED microscope (STEDYCON v.7.1.53 software) mounted on an Olympus IX83 inverted body, equipped with a UPlanXApo (×100/1.45 NA) oil objective lens (Olympus). Samples were excited using 40 MHz pulsed 488 or 640 nm laser lines. Emitted photons were collected by the objective lens, descanned, passed through a 1.0 AU pinhole and recorded by avalanche photodiodes with 500–550 and 580–625 nm filters for 488 nm excitation (photoblued dyes) and a 650–700 nm filter for 640 nm excitation (red dyes). The signal from the avalanche photodiodes was cloned and sent to a Flex02-08D/C correlator card (Correlator.com). FCS data were acquired for 10 s, and for each spot and laser power two measurements were taken in each independent experiment. Between FCS measurements a 20 × 20 μm$^2$ area covering the whole membrane patch to ensure exposition of all fluorescent molecules was irradiated for the specified number of frames. Irradiation settings were: 20 nm pixel size, 10 μs dwell time, one line accumulation and 10 μW excitation laser power at the sample plane. For STED, a pulsed 40 MHz 775 nm laser was used. Laser power was routinely measured using a Thorlabs Power Meter S170C by placing the sensor at the sample stage.

The obtained FCS curves were fitted using the FoCuS point software[24] to a two-dimensional Brownian diffusion model:

$$G_\tau = \frac{1}{N}\left(1 + \left(\frac{\tau}{\tau_D}\right)^\alpha\right)^{-1}$$

where $N$ is the average number of fluorescent particles in the focal volume, $\tau_D$ is the average transient time and $\alpha$ is the anomaly parameter. We included triplet state kinetics for Abberior STAR RED, ATTO 647N and ATTO 655 red detection measurements:

$$G_\tau = \frac{1}{N}\left(1 + \left(\frac{\tau}{\tau_D}\right)^\alpha\right)^{-1}\left(1 - T + T \cdot e^{-\frac{\tau}{\tau_{triplet}}}\right)$$

where $T$ is the fraction that molecules in the triplet state and $\tau_{triplet}$ the triplet time. $\tau_{triplet}$ was fixed to 5 µs for red detection curves, as experimentally determined from independent FCS measurements. Counts per molecule (brightness) were calculated from FCS experiments by dividing the average count rate of the measurement by the average number of molecules in the observation spot ($N$). Molecule percentages were calculated by dividing $N$ by the initial $N$ obtained for the red detection.

To quantify the resolution at different STED powers, POPC SLBs were measured and the apparent spot size was calculated using the following equation[25]:

$$\omega_{0,STED} = \omega_{0,confocal}\sqrt{\frac{\tau_{D(STED)}}{\tau_{D(conf)}}}$$

where $\omega_0$ is the $1/e^2$ radius of the Gaussian beam, and $\tau_{D(STED)}$ and $\tau_{D(conf)}$ are the transit times at a given STED power and in confocal mode, respectively. $\omega_{0,confocal}$ was quantified by performing FCS measurements of ATTO 655 in PBS as a reference dye.

### Fluidity and packing measurements

Measurements were performed in PBS. Two-channel confocal and STED images were acquired in a custom Abberior Expert Line laser scanning STED microscope using a UplanSApo (×100/1.4 NA) oil immersion objective lens (Olympus). NR4A/NR12A were excited by a 561 nm pulsed diode laser PDL-T 561 (Abberior Instruments) with an excitation power of 10 µW at the sample plane. Fluorescence was inhibited by a 775 nm PFL-40-3000-775-B1R 40 MHz pulsed laser (MPB Communications). The STED beam power at the sample plane for each experiment is specified in figure legends. The beam shape for two-dimensional depletion (two-dimensional doughnut) was created using a spatial light modulator (SLM). The STED beam was aligned on top of the excitation point spread function every imaging day using coverslip-immobilized four-color TetraSpeck 100 nm microspheres (Thermo Fisher Scientific, catalog no. T7279) as a reference. The STED beam position was corrected with respect to the confocal signal by adjusting the grating of the SLM. The orientation of the depletion beam was fine-tuned by adjusting the SLM offset. Emitted photons were collected through the objective lens, descanned, passed through a 1.0 AU pinhole, and finally collected by single-photon counting SPCM-AQRH-14-TR avalanche photodiodes (Excelitas Technologies) equipped with appropriate filters (580–630 and 650–700 nm). Pixel size was 40 nm and pixel dwell time was 10 µs. Images were analyzed using a macro (https://doi.org/10.5281/zenodo.5110173) in Fiji that applied an intensity threshold to analyze only pixels corresponding to membranes. Saturated pixels were excluded from the analysis. For every image, the average intensity signal was then calculated for each channel. To quantify membrane packing, the GP ratiometric value for each pixel was calculated using the GP function:

$$GP = \frac{I_b - I_r}{I_b + I_r}$$

where $I_b$ and $I_r$ are the intensity recorded at 580–630 (b) and 650–700 nm (r), respectively. At least three regions of interest were measured for each independent experiment and the mean GP of that region of interest considered.

### FLIM

Measurements were performed in PBS. To deplete oxygen an enzymatic scavenging system was used ('Sample preparation: FCS' section). FLIM data were acquired on the Abberior Instruments Expert Line laser scanning STED microscope described in the 'Fluidity and packing measurements' section. Detected signals were cloned and sent to a time-correlated single-photon-counting Time Tagger device (Swabian Instruments). Lifetime measurements were performed on $5 \times 5$ µm$^2$ regions of interest with a 40 nm pixel size and 10 µs dwell time. Each line was scanned three times. Control measurements of free acid dyes in solution were taken to ensure lifetimes matched those previously reported. Mean pixel lifetimes were analyzed using SPCImage (Becker & Hickl).

### Data analysis and visualization

Fiji v.1.54f was used for image analysis. FoCuS point v.1.13.156 was used for FCS curve fitting. Data analysis and plot preparation were performed in Jupyter Lab v.4.1.5 using Python v.3.9.16 and the following libraries: numpy v.1.24.3, pandas v.2.0.3, matplotlib v.3.7.1 and seaborn v.0.12.2.

### Reporting summary

Further information on research design is available in the Nature Portfolio Reporting Summary linked to this article.

## Data availability

Data and visualization code are available from Zenodo at https://doi.org/10.5281/zenodo.10996577 (ref. 26).

## Code availability

The fluidity analysis macro is available from Zenodo at https://doi.org/10.5281/zenodo.5110173 (ref. 27).

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

## Acknowledgements

This project has received funding from the European Union's Horizon 2020 research and innovation program under the Marie

Skłodowska-Curie grant agreement no. 892232 to P.C. P.C. is supported by KI-KIRI (grant no. 2022-02535). B.K. and I.U. are funded by the Slovenian Research Agency (grant nos. P1-0060, J7-2596). Further financial support by the Deutsche Forschungsgemeinschaft (DFG, the German Research Foundation; under Germany's Excellence Strategy, EXC 2051, Project-ID 390713860; project number 316213987, SFB 1278; GRK M-M-M: GRK 2723/1, 2023, ID 44711651; project PolaRas EG 325/2-1; Instrument funding MINFLUX Jena INST 275_405_1; Instrument funding modular STED INST 1757/25-1 FUGG), the Alexander von Humboldt Foundation (Research Group Linkage Fund), the State of Thuringia (TMWWDG), the Free State of Thuringia (TAB, AdvancedSTED/FGZ, grant no. 2018 FGI 0022; Advanced Flu-Spec/2020 FGZ, grant no. FGI 0031; Multi-XUV/2023 FGR 0054), the innovation program by the German BMWi (ZIM; project no. 16KN070934/Lab-on-a-chip FCS-Easy) and the Leibniz ScienceCampus InfectoOptics Jena (project no. PNEUTHERA, funding line Strategic Networking of the Leibniz Association) to C.E. is greatly acknowledged. Further, this work is supported by the BMBF (The Federal Ministry of Education and Research), funding program LIVE2QMIC (FGZ, grant no. 13N15956) as well as Photonics Research Germany (FKZ, grant nos. 13N15713, 13N15717) and is integrated into the LPI. The LPI is initiated by Leibniz-IPHT, Leibniz-HKI, UKJ and FSU Jena, and is part of the BMBF national roadmap for research infrastructures. We thank F. Schneider and E. Sezgin for constructive comments on the manuscript.

## Author contributions

A.D., A.K. and B.K. performed the experiments. A.D., A.K. and P.C. analyzed the data. B.K. and I.U. supervised spectral STED measurements. A.D. and P.C. conceived the project. C.E. and P.C. supervised the project and acquired funding. C.E. and P.C. wrote the manuscript with contributions from all authors.

## Funding

## Competing interests

The authors declare no competing interests.

## Additional information

**Extended data** is available for this paper at https://doi.org/10.1038/s41592-024-02297-4.

**Correspondence and requests for materials** should be addressed to Christian Eggeling or Pablo Carravilla.

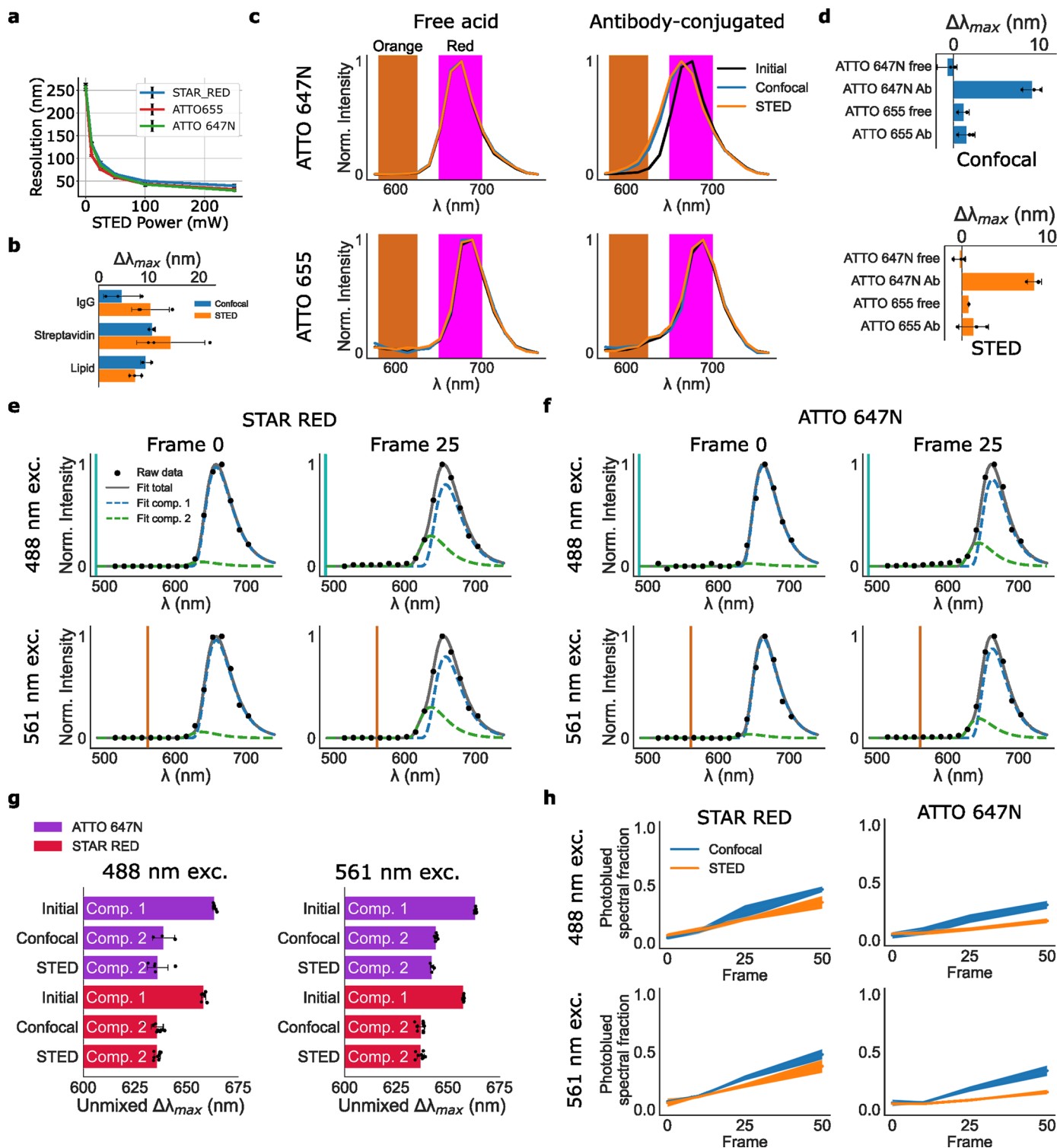

**Extended Data Fig. 1 | See next page for caption.**

**Extended Data Fig. 1 | Characterisation of photoconversion products measured by spectral imaging. a**, STED microscopy spatial resolution (measured as the size ($\omega_0$) of the effective observation spot) obtained for 640 nm excitation and with increasing 775 nm STED laser power as measured by STED-FCS for lipid conjugated Abberior STAR RED, ATTO 647N and ATTO 655 fluorophores diffusing in a SLB. **b**, Changes in fitted emission spectra maxima ($\Delta\lambda_{max}$) after 25 frames of confocal or STED illumination of Abberior STAR RED conjugated to an antibody (IgG), streptavidin and a lipid. For IgG and streptavidin the conjugated molecules were immobilized and imaged in PBS on cover glass (as for Fig. 1b–e) and for the lipid on a SLB patch (as for Fig. 1f–k). Each dot represents an independent measurement. **c**, Conjugation effect on photoblueing susceptibility as measured by spectral imaging. Emission spectrum of free/unconjugated and antibody-conjugated ATTO 647N (upper) and ATTO 655 (lower) on cover glass and in PBS, before (Initial) and after 25 frames of confocal or STED microscopy imaging with 640 nm excitation. Wavelength detection windows following 561 and 640 nm excitation are shown in orange and magenta,

respectively. **d**, Changes in emission spectra maxima ($\Delta\lambda_{max}$) after 25 confocal (top) and STED (bottom) microscopy imaging frames for the two fluorophores and conjugation conditions of panel c. Each point represents one measurement. **e**, **f**, Spectral unmixing of DPPE-conjugated Abberior STAR RED (**e**) and ATTO 647N (**f**) emission spectra (in SLB patches), upon 488 nm (top) or 561 nm (bottom) excitation (vertical lines), before (left) and after (right) 25 frames of confocal irradiation with 640 nm excitation. **g**, Fitted emission spectra maxima ($\Delta\lambda_{max}$) of component 1 (red) and component 2 (photoblued) unmixed spectra of the data from panels **e** and **f** of the DPPE/lipid-conjugated ATTO 647N (purple) and Abberior STAR RED (red). Each point corresponds to one measurement. **h**, Component 2 (photoblued) fitting fraction of spectral imaging data from the data from panels **e** and **f** of the DPPE-conjugated Abberior STAR RED (left) and ATTO 647N (right) upon repeated frames of confocal or STED irradiation with 640 nm excitation. Mean and standard deviation of three independent experiments are shown in **b**, **d**, **g** and **h**.

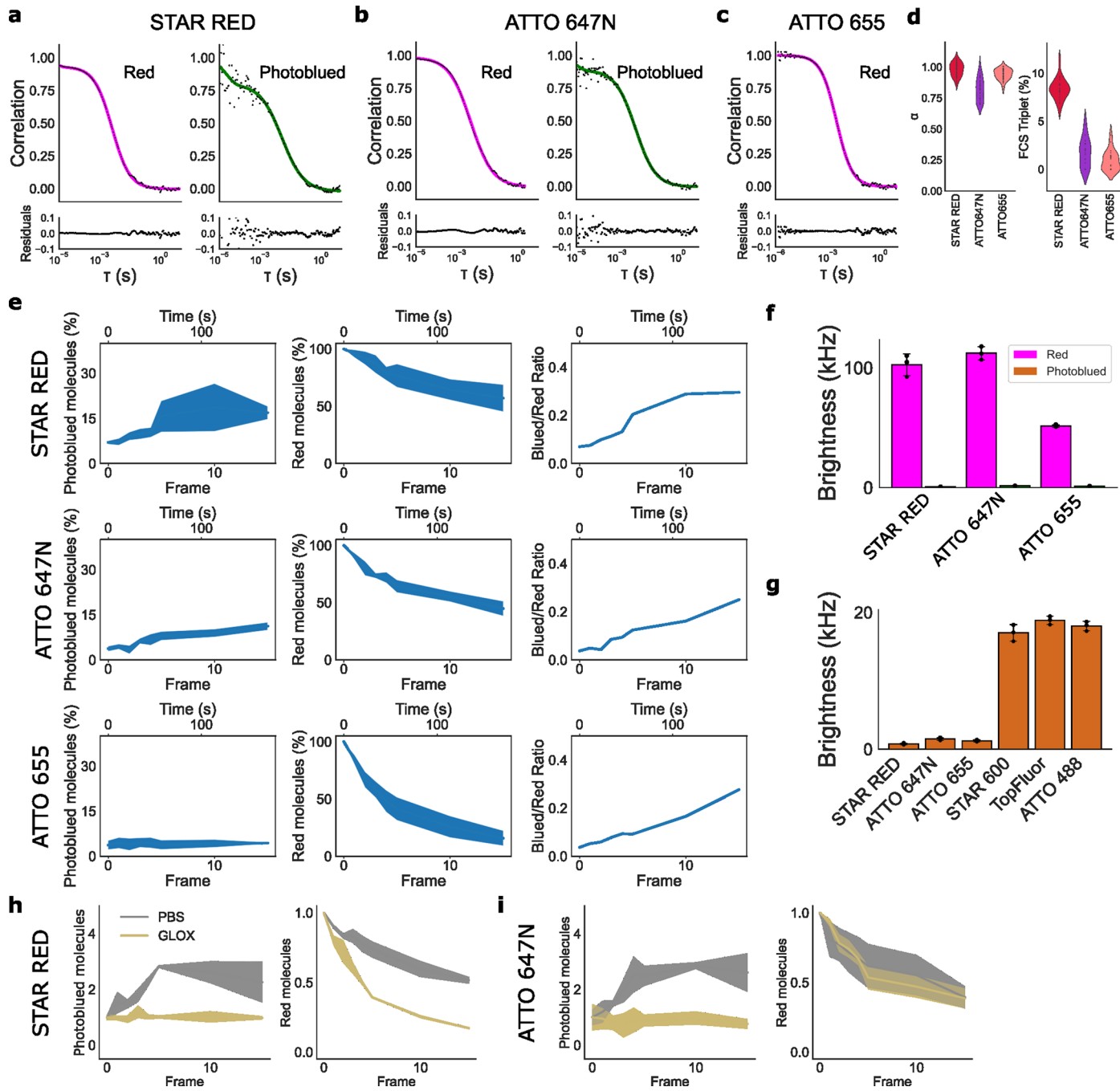

**Extended Data Fig. 2 | Fluorescence correlation spectroscopy analysis of photoconversion products. a-c,** Representative FCS curves, fittings and residuals of Abberior STAR RED (**a**), ATTO 647N (**b**), and ATTO 655 (**c**) conjugated to a lipid/DPPE and diffusing in a SLB patch. Data corresponding to red (left, 640 nm excitation) and photoblued (right, 488 nm excitation) molecules is shown. Photoblueing was induced through 5 imaging frames of the whole SLB patch with 640 nm excitation. Note that ATTO 655 does not undergo photoblueing. **d,** Anomaly parameter (α) and triplet fraction extracted from fitting the FCS data (see *FCS* Methods section). α values close to 1 indicate normal (free/Brownian) diffusion. Note that we observed strongest photoblueing for Abberior STAR RED, followed by ATTO 647N and hardly any for ATTO 655, which correlates with the amount of triplet state population as extracted from the decay in the µs-time range of the FCS data: 8.4 ± 1.0% (Abberior STAR RED), 1.7 ± 1.5% (ATTO 647N) and 0.9 ± 1.0% (ATTO 655). Each point represents an FCS

measurement. **e,** FCS measurements of molecule number N (relative to time point 0) of the fluorescent lipid conjugates in the SLB patches as highlighted in Fig. 1f–k show that photoconversion (combination of photobleaching and generation of blue fluorescence) increases the ratio of blued-to-red dyes up to 30% in all cases. **f, g,** Fluorescence brightness (counts per molecule) of photoblued dyes (488 nm excitation) as determined from the FCS experiments of Fig. 1f–k is orders of magnitude lower than that of the original red counterparts (top, 640 nm excitation) and popular green and orange dyes (bottom, 488 nm excitation). Each point represents the average of all measurements of an independent experiment. **h, i,** Normalised number of photoblued (left) and red (right) molecules as measured in PBS and in an enzymatic oxygen scavenging buffer (GLOX) for Abberior STAR RED (**h**) and ATTO 647N (**i**). All measurements are performed with DPPE-conjugated dyes in POPC SLBs. Mean and standard deviation of three independent experiments are shown in **e-i**.

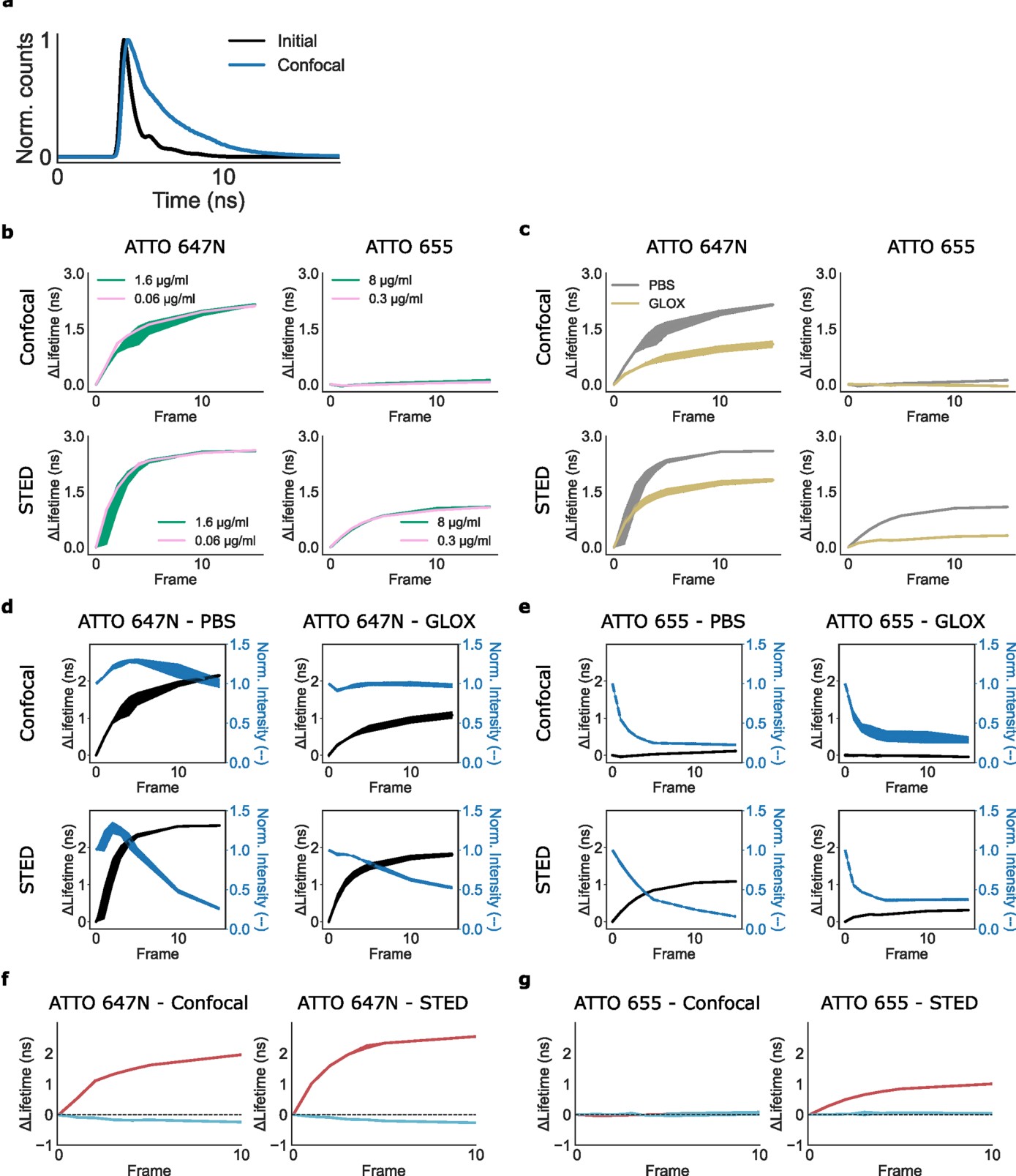

**Extended Data Fig. 3 | Photoconversion and fluorescence lifetime.**
**a**, Representative fluorescence lifetime curves of antibody (IgG)-conjugated ATTO 647N on cover glass and in PBS before and after 10 confocal microscopy imaging frames with 640 nm excitation. **b**, Lifetime changes reported in Fig. 1l–n are concentration-independent. **c**, Lifetime changes of IgG-conjugated ATTO 647N (left) and ATTO 655 (right) measured in PBS and in an enzymatic oxygen scavenging buffer (GLOX), upon repeated frames of confocal (top) or

STED (bottom) illumination. **d-e**, Simultaneous quantification of mean intensity (dashed line) and lifetime changes of IgG-conjugated ATTO 647N (**d**) and ATTO 655 (**e**), upon confocal (top) and STED (bottom) irradiation with 640 nm and 640/775 nm, respectively, as measured in PBS and GLOX. **f-g**, Comparison of lifetime changes of IgG- and DPPE-conjugated ATTO 647N (**f**) and ATTO 655 upon confocal (left) and STED (right) irradiation. Mean and standard deviation are shown in **b-g**.

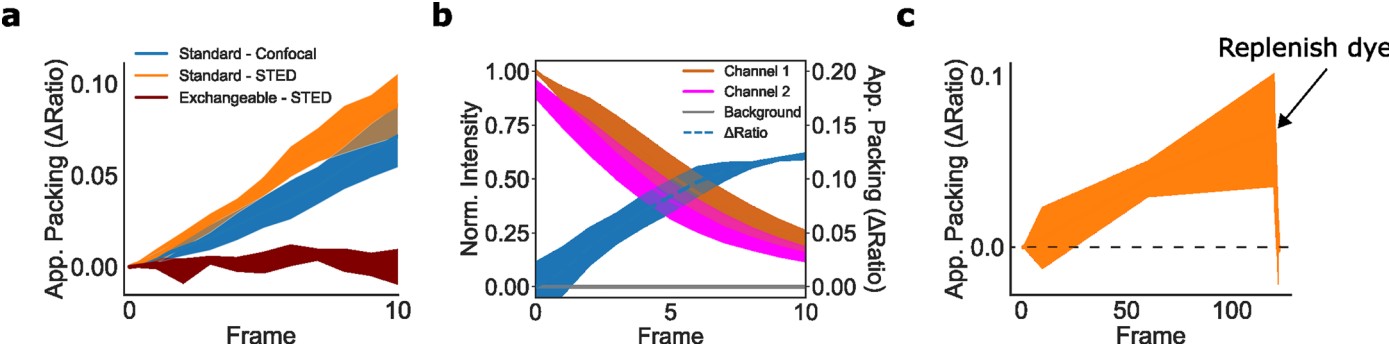

**Extended Data Fig. 4 | Bias in membrane lipid packing measurements is due to photoblueing. a**, Changes in the apparent membrane lipid packing as determined from the signal GP ratio of the environment-sensitive fluorophores NR12A (Standard) and NR4A (Exchangeable) in POPC SLBs membrane patches upon confocal and STED microscopy imaging (640 nm excitation, 775 nm STED) for repeated imaging frames. **b**, Intensity changes for channel 1 (580–630 nm), channel 2 (650–700 nm), and background intensity and subsequent changes in apparent packing as measured for NR12A in POPC SLBs membrane patches. **c**, Measurements as in **a** but with unbiased membrane lipid packing values recovered upon addition of fresh dye to the sample after several frames of STED microscopy imaging. Mean and standard deviation of three independent experiments are shown.

Pablo Carravilla

# Reporting Summary

## Statistics

For all statistical analyses, confirm that the following items are present in the figure legend, table legend, main text, or Methods section.

| n/a | Confirmed | |
|---|---|---|
| ☐ | ☒ | The exact sample size (*n*) for each experimental group/condition, given as a discrete number and unit of measurement |
| ☐ | ☒ | A statement on whether measurements were taken from distinct samples or whether the same sample was measured repeatedly |
| ☒ | ☐ | The statistical test(s) used AND whether they are one- or two-sided<br>*Only common tests should be described solely by name; describe more complex techniques in the Methods section.* |
| ☒ | ☐ | A description of all covariates tested |
| ☒ | ☐ | A description of any assumptions or corrections, such as tests of normality and adjustment for multiple comparisons |
| ☐ | ☒ | A full description of the statistical parameters including central tendency (e.g. means) or other basic estimates (e.g. regression coefficient) AND variation (e.g. standard deviation) or associated estimates of uncertainty (e.g. confidence intervals) |
| ☒ | ☐ | For null hypothesis testing, the test statistic (e.g. *F*, *t*, *r*) with confidence intervals, effect sizes, degrees of freedom and *P* value noted<br>*Give P values as exact values whenever suitable.* |
| ☒ | ☐ | For Bayesian analysis, information on the choice of priors and Markov chain Monte Carlo settings |
| ☒ | ☐ | For hierarchical and complex designs, identification of the appropriate level for tests and full reporting of outcomes |
| ☒ | ☐ | Estimates of effect sizes (e.g. Cohen's *d*, Pearson's *r*), indicating how they were calculated |

*Our web collection on statistics for biologists contains articles on many of the points above.*

## Software and code

Policy information about availability of computer code

| Data collection | Data were acquired using Inspector 16.3.15517 (Abberior Expert Line microscope) and STEDYCON 7.1.53 (Abberior STEDYCON microscope). |
|---|---|
| Data analysis | Fiji 1.54f was used for image analysis (https://fiji.sc). FoCuS point 1.13.156 was used for FCS curve fitting. Data analysis and plot preparation were performed in Jupyter Lab 4.1.5 using Python 3.9.16 and the following libraries: numpy 1.24.3, pandas 2.0.3, matplotlib 3.7.1 and seaborn 0.12.2. |

For manuscripts utilizing custom algorithms or software that are central to the research but not yet described in published literature, software must be made available to editors and reviewers. We strongly encourage code deposition in a community repository (e.g. GitHub). See the Nature Portfolio guidelines for submitting code & software for further information.

## Data

Policy information about availability of data

All manuscripts must include a data availability statement. This statement should provide the following information, where applicable:
- Accession codes, unique identifiers, or web links for publicly available datasets
- A description of any restrictions on data availability
- For clinical datasets or third party data, please ensure that the statement adheres to our policy

Data and visualisation code are available from Zenodo (https://zenodo.org/doi/10.5281/zenodo.10996577).

## Human research participants

Policy information about studies involving human research participants and Sex and Gender in Research.

| | |
|---|---|
| Reporting on sex and gender | N/A |
| Population characteristics | N/A |
| Recruitment | N/A |
| Ethics oversight | N/A |

Note that full information on the approval of the study protocol must also be provided in the manuscript.

# Field-specific reporting

Please select the one below that is the best fit for your research. If you are not sure, read the appropriate sections before making your selection.

☒ Life sciences    ☐ Behavioural & social sciences    ☐ Ecological, evolutionary & environmental sciences

For a reference copy of the document with all sections, see nature.com/documents/nr-reporting-summary-flat.pdf

# Life sciences study design

All studies must disclose on these points even when the disclosure is negative.

| | |
|---|---|
| Sample size | No statistical method was used for sample size determination. Three independent replicates of each experiment were performed (n=3). This sample size was resulted in low data variability (measured as the standard deviation of the independent replicates) and was considered sufficient to accurately determine the photophysical parameteres measured in this study: emission spectrum, average number of molecules, molecular brightness, average fluorescence lifetime and generalised polarisation. |
| Data exclusions | No data were excluded. |
| Replication | Three independent replicates were performed for each condition. To ensure reproducibility average values of independent experiments are considered, not individual measurements. All replication attempts were successful. |
| Randomization | No randomization measures. Randomization was not deemed necessary, as the results of our experiments are quantitatively measurable parameters that are not influenced by subjective bias. |
| Blinding | No blinding. It was not deemed necessary, as the results of our experiments are quantitatively measurable parameters that are not influenced by subjective bias. |

# Reporting for specific materials, systems and methods

We require information from authors about some types of materials, experimental systems and methods used in many studies. Here, indicate whether each material, system or method listed is relevant to your study. If you are not sure if a list item applies to your research, read the appropriate section before selecting a response.

## Materials & experimental systems

| n/a | Involved in the study |
|---|---|
| ☐ | ☒ Antibodies |
| ☐ | ☒ Eukaryotic cell lines |
| ☒ | ☐ Palaeontology and archaeology |
| ☒ | ☐ Animals and other organisms |
| ☒ | ☐ Clinical data |
| ☒ | ☐ Dual use research of concern |

## Methods

| n/a | Involved in the study |
|---|---|
| ☒ | ☐ ChIP-seq |
| ☒ | ☐ Flow cytometry |
| ☒ | ☐ MRI-based neuroimaging |

## Antibodies

| | |
|---|---|
| Antibodies used | The following antibodies were used: |

| Antibodies used | Abberior STAR RED goat anti-mouse IgG (Abberior, STRED-1001), Anti-Mouse-IgG - Atto 647N antibody produced in goat (Sigma-Aldrich, 50185), Anti-Mouse IgG - Atto 655 antibody produced in goat (Sigma-Aldrich, 50283) |
|---|---|
| Validation | No validation was required as antibodies were not used for epitope recognition. |

# Eukaryotic cell lines

Policy information about cell lines and Sex and Gender in Research

| Cell line source(s) | PtK2 (NBL-5) were acquired from ATCC (CCL-56) |
|---|---|
| Authentication | None |
| Mycoplasma contamination | Cells tested negative for mycoplasma. |
| Commonly misidentified lines (See ICLAC register) | No commonly misidentified cell lines were used in this study. |

