## [Peer Review File · Nature Methods]

Peer Review Information

Manuscript Title: Effects and avoidance of photoconversion-induced artefacts in confocal and STED microscopy

Corresponding author name(s): Christian Eggeling, Pablo Carravilla

Editorial Notes: None

Reviewer Comments & Decisions:

Decision Letter, initial version:

Dear Pablo,

Your Brief Communication, "Effects and avoidance of photoconversion-induced artefacts in confocal and STED microscopy", has now been seen by three reviewers. As you will see from their comments below, although the reviewers find your work of considerable potential interest, they have raised a number of concerns. We are interested in the possibility of publishing your paper in Nature Methods, but would like to consider your response to these concerns before we reach a final decision on publication. We therefore invite you to revise your manuscript to address these concerns.

To our view, the referee concerns and comments were reasonable and constructive, and we think addressing them is necessary to complete the story and better understand the mechanistic underpinnings of the observations. We hope the revision can make it very clear what (if any) observations are caused by selective photobleaching or other phenomena rather than photobleaching.

We are committed to providing a fair and constructive peer-review process. Do not hesitate to contact us if there are specific requests from the reviewers that you believe are technically impossible or unlikely to yield a meaningful outcome. In particular, please let us know if you find the request for mass spec analysis made by reviewer 1 unfeasible.

- * include a point-by-point response to the reviewers and to any editorial suggestions
- * please underline/highlight any additions to the text or areas with other significant changes to facilitate review of the revised manuscript
- * address the points listed described below to conform to our open science requirements

* ensure it complies with our general format requirements as set out in our guide to authors at www.nature.com/naturemethods

* resubmit all the necessary files electronically by using the link below to access your home page

[Redacted]

We hope to receive your revised paper within three months. If you cannot send it within this time, please let us know. In this event, we will still be happy to reconsider your paper at a later date so long as nothing similar has been accepted for publication at Nature Methods or published elsewhere.

OPEN SCIENCE REQUIREMENTS

REPORTING SUMMARY AND EDITORIAL POLICY CHECKLISTS

IMAGE INTEGRITY

When submitting the revised version of your manuscript, please pay close attention to our Digital Image Integrity Guidelines and to the following points below:

-- that unprocessed scans are clearly labelled and match the gels and western blots presented in

figures.

- that control panels for gels and western blots are appropriately described as loading on sample processing controls
- all images in the paper are checked for duplication of panels and for splicing of gel lanes.

DATA AVAILABILITY

All novel DNA and RNA sequencing data, protein sequences, genetic polymorphisms, linked genotype and phenotype data, gene expression data, macromolecular structures, and proteomics data must be deposited in a publicly accessible database, and accession codes and associated hyperlinks must be provided in the "Data Availability" section.

MATERIALS AVAILABILITY

As a condition of publication in Nature Methods, authors are required to make unique materials

promptly available to others without undue qualifications.

ORCID

Sincerely,
Rita

Rita Strack, Ph.D.
Senior Editor
Nature Methods

Reviewers' Comments:

Reviewer #1:

Remarks to the Author:

The attached manuscript describe the impact and scope of photobleaching reaction under STED conditions. The manuscript first illustrates the potential for photobleaching of various antibody conjugates and the observation that Atto 655 does not undergo this process. The authors then carry out FLIM based measurements which reveal an extension of the lifetime in irradiation dependent manner, though the link of this observation to photobleaching isn't very clear (see below). Finally, I think the most significant observation is that non-exchangeable membrane dyes can show light-dependent artifact that appears to clearly due to photobleaching. However, the corresponding exchangeable variants do not show this phenomenon. Overall, this really interesting study with a range of important implications.

1. The clearest chemical basis for these observations is an N-dealkylation pathway (DOI: 10.1021/jacs.8b11036 and 10.1021/acs.jpca.9b03588). The authors might speculate (perhaps in supplementary figure) on the possible role of this chemistry in this work. It would be interesting, but perhaps beyond the scope of this work, to carry out simple MS experiments to assess if these probes differentially undergo this chemistry. This is especially feasible on the Nile Red derivatives and Atto-647, where the probes are broadly available, and the structures are well known.

2. Given the work above, the role of oxygen in the photobleaching process should be investigated, given prior results. I also had trouble finding exactly what buffers (just PBS?) the different experiments are run in. The issue of buffer and oxygen effects should be addressed carefully.

3. The issue of the fluorescence lifetime lengthening in the context of antibody labeling may arise from a completely distinct phenomenon. Various groups have reported that particularly protein conjugates get brighter during initial stages of an imaging experiment (for example, 10.1021/acscentsci.1c00483). This is generally attributed to partial bleaching (that is complete loss of chromophore through an oxidative pathway), which in turn disrupts dye aggregates leading to a bright protein conjugate. I suspect the same thing is going on here, since of course lifetime and brightness are often linked properties. This could be easily tested by looking at the lifetime effects of the membrane dyes or even lifetime effects of isolated antibody conjugates vs. the corresponding well solubilized probe. Before the FLIM data is included, this issue should be clarified, since in my mind just bleaching-related properties are distinct from photobleaching effects.

4. This is a minor point, but the word PAINT refers to the acronym "points accumulation for imaging in nanoscale topography", but since STED is an ensemble and not a single molecule method, no point signals are being accumulated in this case. I would think the word "exchangeable" (as used in the abstract) better describes the experiments being run in the membrane section.

Martin Schnermann
National Cancer Institute

Reviewer #2:

Remarks to the Author:

The manuscript by Dasgupta et al describes the problem of photobleaching in confocal and STED microscopy and proposes the method to avoid it. The photobleaching phenomenon is demonstrated on multiple fluorophores, which can impact both the emission color and lifetime. Then, it is shown that an exchangeable dye can resolve this problem because of continuous renewal of the dye. The originality and significance of this work is high, as it demonstrates that photobleaching is a rather general phenomenon impacting a broad variety of dyes and experimental conditions. Moreover, the avoidance of this phenomenon due to dye exchange can stimulate development of new methods that do not suffer from the photobleaching problem. The data are clearly presented and the conclusions are generally supported by the data. However, several issues should be addressed before this manuscript can be recommended for publication in Nature Methods.

1) Generally, the photobleaching phenomena reported to date lead to drastic changes in the absorption

and emission spectra. For example, in the work by Sauer and co-workers (ref. 9), the photoconversion in a cyanine-5 and Atto647N dyes led to >100 nm blue shift in the absorption/emission. In the present case, the changes observed for all dyes were much smaller and were generally limited to 10-20 nm, accompanied by some band broadening (Figure 1c,d, and 2a). First, one should clarify better where the emission band maximum of photoblued species appear. Could spectral deconvolution help here? The other question is whether the authors really observe a photoconversion or some effects of environmental factors. For example, strong illumination could selectively bleach red-shifted fluorophores, rather than blue shifted ones. Moreover, it is shown that photoblueing is observed only when the dye is grafted to the antibody, but not for the free dye in solution. This suggests that the factors of the probe environment could influence the phenomenon. It would be useful to check whether STAR RED (used in this work) also shows this difference between grafted and free form. In case of photoconversion (e.g. in ref. 9), a break or change in the conjugation would lead to dramatic change in the emission color. Therefore, much more fine changes should take place in the dyes to observe this phenomenon, like N-dealkylation (ref. 4). Moreover, most common mechanisms of photobleaching are related to photooxidation. Therefore, I would suggest making photoblueing experiments in deoxygenated conditions. Overall, the authors should provide a discussion on possible mechanisms of the observed photoblueing.

2) I have a similar question to the data on NR12A in Figure 2a and extended data Fig. 2: how environmental factors could affect photoblueing of this probe? The cell membrane is microheterogeneous, which could generate dye species with blue and red shifted emission maxima. Then, photobleaching could bleach preferentially red species compared to the blue ones, producing apparent photoblueing. The experiments with replenishment of the dye cannot completely exclude the environmental factors, because addition of the fresh dye should drastically increase the fluorescence intensity because the remaining species after photobleaching should be rather poorly emissive. In this respect, the information about the changes in the fluorescence intensity for both detection channels should be provided during the photobleaching (for each frame). One approach to verify the effect of eventual heterogeneity in the membranes is to test photoblueing phenomenon for NR12A in POPC supported lipid bilayers (as it was done for other dyes, Figure 1), which is expected to be much more homogeneous than the cell membrane.

3) The authors used 561 nm (vs 640 nm) excitation to show the photoblueing in Figure 1b, while the changes in the shifts in the emission are only 10-20 nm. In case of the supported lipid bilayers the excitation was done at even shorter wavelength (488 nm). This raises a question whether the authors really observe the same type of photoblueing in figures 1b and 1c,d. To clarify this, fluorescence spectra at 561 excitation should be also recorded before and after photoblueing (like it is done for Figure 1c,d). This could also help to identify better the emission maximum of the photoblued species. Moreover, the emission spectra for the studied lipid conjugates in supported bilayers before and after photobleaching should be also shown for both 640 and 488 nm excitation.

4) The use of term "sensor" usually stands for devices. Term "probe" would be more appropriate here.

Reviewer #3:

Remarks to the Author:

The paper by Dasgupta and co-authors investigates photoconversion induced artefacts in fluorescence microscopy. It is a short paper but it packs a lot in and makes a number of important points. With

fluorescence microscopy becoming more and more turnkey and procedure based, it is timely to be reminded of the potential for imaging artefacts and the need to be aware to the possibility of these in any given imaging experiment.

The authors make a good selection of dye molecules for study: Atto 647N, Star red, and Atto 655, which are all commonly used dyes, although they are all red emitters and it would have been a nice inclusion to have another dye or two from a different part of the spectrum.

Even so, from the three dyes studied, some interesting results are obtained, especially around the phenomenon of photobleaching. Firstly, the photoconversion spectra of the dyes clearly show how emission from photobleached molecules has the potential to contribute to other detection channels, and in this context, it is good to know that the photobleaching products are dim. Secondly, it is interesting that Atto 655 does not photobleach. Can the authors comment on why it is resistant to this, and why the other two dyes are susceptible to photobleaching? And related to this, do the authors have any thoughts on why there is a greater effect on dyes conjugated to antibodies?

The points about FLIM and FRET and changes in lifetimes are well made. Often a small change in lifetime, such as that seen for Atto 647N, is interpreted as a meaningful result in terms of the experiment being undertaken but the observation that it could arise from photobleaching is important to know.

In the same vein, the observation of spectral shifts of a few nanometres are also often interpreted as a result and the authors clearly demonstrate that such shifts can arise from photobleaching. Restoring the dye to prove that the observed shifts are an artefact and not the result of packing changes in a membrane is an excellent demonstration of this point. Establishment of the Nile red derivative NR4A as a suitable dye for these applications is also a useful contribution.

Some example FCS curves and fits should be included in the extended data along with fitting results, especially the anomaly parameter values. Also, why was 5 μ s chosen for the triplet time for all dyes in FCS fitting? Aren't these values known or couldn't this component be allowed to be freely fit?

It would be useful to note in the text where a number of frames observation period is noted, what this is in terms of time, to save having to work this out from the experimental details. E.g. lines 41, 56, 66.

Author Rebuttal to Initial comments

Reviewer #1:

Remarks to the Author:

The attached manuscript describe the impact and scope of photobleaching reaction under STED conditions. The manuscript first illustrates the potential for photobleaching of various antibody conjugates and the observation that Atto 655 does not undergo this process. The authors then carry out FLIM based measurements which reveal an extension of the lifetime in irradiation dependent manner, though the link of this observation to photobleaching isn't very clear (see below). Finally, I think the most significant observation is that non-exchangeable membrane dyes can show light-dependent artifact that appears to clearly due to photobleaching. However, the corresponding exchangeable variants do not show this phenomenon. Overall, this really interesting study with a range of important implications.

We thank the reviewer for his positive assessment and insightful comments that helped us improve the manuscript. We believe the current version addressed all the raised points. In particular, we could now expand our interpretation of the FLIM data and its link to different photoreaction products, as pointed out below.

1. The clearest chemical basis for these observations is an N-dealkylation pathway (DOI: 10.1021/jacs.8b11036 and 10.1021/acs.jpca.9b03588). The authors might speculate (perhaps in supplementary figure) on the possible role of this chemistry in this work. It would be interesting, but perhaps beyond the scope of this work, to carry out simple MS experiments to assess if these probes differentially undergo this chemistry. This is especially feasible on the Nile Red derivatives and Atto-647, where the probes are broadly available, and the structures are well known.

We agree with the reviewer that MS experiments would shed light on the chemistry underlying our observations, and we have indeed thought about these. However, for us they would require an experimental implementation with a long time scale that is beyond the scope of the present study, and we would leave them for future studies. Nevertheless, we performed new experiments on photobleaching in the absence of oxygen as suggested by the reviewer, which gave us some new insights into possible chemical reactions involved (see next comment).

2. Given the work above, the role of oxygen in the photobleaching process should be investigated, given prior results. I also had trouble finding exactly what buffers (just PBS?) the different experiments are run in. The issue of buffer and oxygen effects should be addressed carefully.

Following the reviewer's suggestion, we quantified photobleaching and photobleaching in deoxygenated conditions (new Extended Data Fig. 2i-j) and observed a drastic decrease in the photobleaching rate. This suggests that photobleaching involves photooxidative reactions as reported in the mentioned papers.

We also assessed the effect of oxygen in lifetime changes (new Extended Data Fig. 3c), which affected both ATTO 655 and ATTO 647N moderately (see comment 3).

We now discuss the potential role of different photobleaching pathways via the triplet state and through interaction with reactive oxygen:

"To explore possible mechanisms of the photobleaching pathway, we investigated dependencies on oxygen. Oxygen is known to have two major effects on the dyes' photophysics: 1) quenching of the triplet state and therefore less photoreactions out of that state, and 2) increased general photoreactivity due to interaction with the dissolved oxygen, especially when in its singlet state^{2,10}. We consistently observed that, photobleaching was suppressed as a photoreaction in the absence of oxygen (Extended Data Fig. 2i,j), as highlighted before for other dyes^{6,9}. This on one hand indicates that the photobleaching pathway involve photooxidation (such as in photooxidative N-dealkylation processes¹⁰ or through singlet oxygen produced by quenching of the dyes' triplet state¹⁰) and hardly evolves directly through the triplet state, and on the other hand that photobleaching is only one of many possible photoreaction pathways. Interestingly, our FCS data highlights a correlation between the extend of photobleaching and the amount of intersystem crossing and thus triplet yield, being lowest for ATTO 655 and largest for Abberior STAR RED (Extended Data Fig. 2d). Since singlet oxygen is generated through triplet-triplet energy transfer, we speculate that photobleaching is fostered via this mechanism."

We also consider the differential effect of oxygen in photobleaching and lifetime changes, which suggest that these result from different photoreaction products:

“This fact and the observation that the lifetime change was only partially inhibited upon enzymatic oxygen depletion (Extended Data Fig. 3c), highlighted that lifetime changes are not a consequence of photobleaching but rather due to the outlined partial photobleaching.”

Last, we updated the methods and figure legends to clearly explain all the employed buffers.

3. The issue of the fluorescence lifetime lengthening in the context of antibody labeling may arise from a completely distinct phenomenon. Various groups have reported that particularly protein conjugates get brighter during initial stages of an imaging experiments (for example, 10.1021/acscentsci.1c00483). This is generally attributed to partial bleaching (that is complete loss of chromophore through an oxidative pathway), which in turn disrupts dye aggregates leading to a bright net protein conjugate. I suspect the same thing is going on here, since of course lifetime and brightness are often linked properties. This could be easily tested by looking at the lifetime effects of the membrane dyes or even lifetimes effects of isolated antibody conjugates vs. the corresponding well solubilized probe. Before the FLIM data is included, this issue should be clarified, since in my mind just bleaching-related properties are distinct from photobleaching effects.

We thank the reviewer for this important comment that allowed us to better understand the source of lifetime changes.

To assess the effect of partial bleaching, we quantified intensity changes during FLIM experiments in the presence and absence of oxygen. We did indeed observe an overall intensity increase during the initial frames for ATTO 647N, which corresponded to the sharpest lifetime change (new Extended Data Fig. 3d). This suggests that lifetime changes can be linked to partial bleaching. To minimise the potential effect of antibody aggregates, we repeated FLIM experiments using 25 times lower concentration and observed a comparable lifetime increase (see new Extended Data Fig. 3b), suggesting that intermolecular aggregates do not explain the observed effect.

Interestingly, as hinted by the reviewer, we did not observe a lifetime increase when measuring lipid conjugated dyes, which contain a single fluorophore per molecule. Instead, we observed a slight decrease (250 ps) of lifetime upon irradiation for ATTO 647N and no changes for ATTO 655 (new Extended Data Fig. 3f-g).

We now in detail describe the possible reasons for lifetime changes in the text and clarified that lifetime changes and photobleaching derive from different processes:

“These data were recorded on antibody-conjugated dyes. Here, a known phenomenon arises from high labelling degrees, i.e. of more than one dye conjugated to a single antibody, which results in quenching effects and thus fluorescence lifetime changes of the dyes conjugated in close proximity^{12,13}. Illumination will in this case result in an initial partial bleaching of one of the dyes and thus a rise in lifetime and an initial rise in fluorescence intensity followed by the drop due to photobleaching of all dyes, as observed in our case (Extended Data Fig. 3d,e). This fact and the observation that the lifetime change was only partially inhibited upon enzymatic oxygen depletion (Extended Data Fig. 3c), highlighted that lifetime changes are not a consequence of photobleaching but rather due to the outlined partial photobleaching. Consistently, we did not observe a lifetime increase upon illumination for lipids conjugated to single dyes, which instead showed a minor lifetime decrease (Extended Data Fig. 3f). Consequently, single-dye conjugation will serve as an artefact mitigation strategy in FLIM experiments.”

Overall, we believe that our data is relevant to researchers performing FLIM experiments, as antibody conjugates are commonly employed. Our data suggest that FLIM measurements using antibodies conjugated to multiple dyes can be affected by artefactual lifetime increases derived from partial bleaching, which can be mitigated by single-dye conjugation strategies.

4. This is a minor point, but the word PAINT refers to the acronym “points accumulation for imaging in nanoscale topography”, but since STED is an ensemble and not a single molecule method, no point signals are being accumulation in this case. I would think the word “exchangeable” (as used in the abstract) better describes the experiments being run in the membrane section.

We agree and changed the text and figure labels accordingly.

Martin Schnermann
National Cancer Institute

Reviewer #2:

Remarks to the Author:

The manuscript by Dasgupta et al describes the problem of photobleaching in confocal and STED microscopy and proposes the method to avoid it. The photobleaching phenomenon is demonstrated on multiple fluorophores, which can impact both the emission color and lifetime. Then, it is shown that an exchangeable dye can resolve this problem because of continuous renewal of the dye. The originality and significance of this work is high, as it demonstrates that photobleaching is a rather general phenomenon impacting a broad variety of dyes and experimental conditions. Moreover, the avoidance of this phenomenon due to dye exchange can stimulate development of new methods that do not suffer from the photobleaching problem. The data are clearly presented and the conclusions are generally supported by the data. However, several issues should be addressed before this manuscript can be recommended for publication in Nature Methods.

We thank the reviewer for their positive evaluation of our work. Following their suggestion, we performed several new experiments. We believe that in this updated version we clarified all the raised issues.

1) Generally, the photobleaching phenomena reported to date lead to drastic changes in the absorption and emission spectra. For example, in the work by Sauer and co-workers (ref. 9), the photoconversion in a cyanine-5 and Atto647N dyes led to >100 nm blue shift in the absorption/emission. In the present case, the changes observed for all dyes were much smaller and were generally limited to 10-20 nm, accompanied by some band broadening (Figure 1c,d, and 2a). First, one should clarify better where the emission band maximum of photobleached species appear. Could spectral deconvolution help here?

Following this valuable suggestion, we performed spectral unmixing to determine the emission band of the photobleached species upon 488 nm and 561 nm excitation (new Extended Data Fig. 1e-g). This strategy allowed us to identify the emission maximum of photobleached ATTO 647N (643.6 ± 1.2 nm) and STAR RED (637.9 ± 1.6 nm), and to calculate the emission fraction corresponding to the photobleached species (new Extended Data Fig. 1h). These spectral changes are indeed smaller than those observed by Sauer and co-workers, but enough to leak into emission ranges used for orange dyes (Extended Data Fig. 1c). We have added our new experiments as well as a comment to ref 9 in the manuscript.

The other question is whether the authors really observe a photoconversion or some effects of environmental factors. For example, strong illumination could selectively bleach red-shifted fluorophores, rather than blue shifted ones. Moreover, it is shown that photobleaching is observed only when the dye is grafted to the antibody, but not for the free dye in solution. This suggests that the factors of the probe environment could influence the phenomenon. It would be useful to check whether STAR RED (used in this work) also shows this difference between grafted and free form. In case of photoconversion (e.g. in ref. 9), a break or change in the conjugation would lead to dramatic change in the emission color. Therefore, much more fine changes should take place in the dyes to observe this phenomenon, like N-dealkylation (ref. 4). Moreover, most common mechanisms of photobleaching are related to photooxidation. Therefore, I would suggest making photobleaching experiments in deoxygenated conditions. Overall, the authors should provide a discussion on possible mechanisms of the observed photobleaching.

To assess the environmental effect, we tried measuring spectral changes of the free version of STAR RED, as suggested by the reviewer. However, due to the high hydrophilicity of the dye, we were not able to immobilise it on

the glass substrate. Alternatively, we performed spectral imaging of a streptavidin-conjugated version (immobilised) and a lipid-conjugated version (diffusive) of STAR RED, which showed a comparable blue shift to the one observed for the antibody (new Extended Data Fig. 1b).

Prompted by this reviewer and reviewer 1, we repeated key experiments in deoxygenated conditions. We observed that photobleaching was almost fully prevented in the absence of oxygen (new Extended Data Fig. 2i-j). Interestingly, lifetime changes were not fully prevented in the case of ATTO 647N (new Extended Data Fig. 3c-e), suggesting that photobleaching and lifetime changes occur as a consequence of distinct processes.

We now discuss potential mechanisms under the light of this new data and in the context of previously published works, including N-dealkylation. We also discuss the possibility that red illumination could preferentially bleach red-shifted fluorophores, as photobleached species are probably excited less efficiently upon 640 nm excitation, increasing the relative number of photobleached molecules (Extended Data Fig. 2e):

“To explore possible mechanisms of the photobleaching pathway, we investigated dependencies on oxygen. Oxygen is known to have two major effects on the dyes’ photophysics: 1) quenching of the triplet state and therefore less photoreactions out of that state, and 2) increased general photoreactivity due to interaction with the solved oxygen, especially when in its singlet state^{2,10}. We consistently observed that, photobleaching was suppressed as a photoreaction in the absence of oxygen (Extended Data Fig. 2i,j), as highlighted before for other dyes^{6,9}. This on one hand indicates that the photobleaching pathway involve photooxidation (such as in photooxidative N-dealkylation processes¹⁰ or through singlet oxygen produced by quenching of the dyes’ triplet state¹⁰) and hardly evolves directly through the triplet state, and on the other hand that photobleaching is only one of many possible photoreaction pathways. Interestingly, our FCS data highlights a correlation between the extend of photobleaching and the amount of intersystem crossing and thus triplet yield, being lowest for ATTO 655 and largest for Abberior STAR RED (Extended Data Fig. 2d). Since singlet oxygen is generated through triplet-triplet energy transfer, we speculate that photobleaching is fostered via this mechanism.”

and

“This fact and the observation that the lifetime change was only partially inhibited upon enzymatic oxygen depletion (Extended Data Fig. 3c), highlighted that lifetime changes are not a consequence of photobleaching but rather due to the outlined partial photobleaching.”

and

“The photobleached species were less photobleached by the 640 nm light than the original red-emitting dyes.”

2) I have a similar question to the data on NR12A in Figure 2a and extended data Fig. 2: how environmental factors could affect photobleaching of this probe? The cell membrane is microheterogeneous, which could generate dye species with blue and red shifted emission maxima. Then, photobleaching could bleach preferentially red species compared to the blue ones, producing apparent photobleaching. The experiments with replenishment of the dye cannot completely exclude the environmental factors, because addition of the fresh dye should drastically increase the fluorescence intensity because the remaining species after photobleaching should be rather poorly emissive. In this respect, the information about the changes in the fluorescence intensity for both detection channels should be provided during the photobleaching (for each frame). One approach to verify the effect of eventual heterogeneity in the membranes is to test photobleaching phenomenon for NR12A in POPC supported lipid bilayers (as it was done for other dyes, Figure 1), which is expected to be much more homogenous than the cell membrane.

We thank the reviewer for this valuable suggestion. In the updated version we include the changes in intensity for each detection channel, where we observe no significant bleaching preference of red-shifted probes (Extended Data Fig. 4b).

In the original manuscript, we did indeed include the suggested POPC control, but mislabelled it as cell data (Extended Data Fig. 4). We now corrected this and updated this figure to include the exchangeable dye control performed in model membranes.

3) The authors used 561 nm (vs 640 nm) excitation to show the photobleuing in Figure 1b, while the changes in the shifts in the emission are only 10-20 nm. In case of the supported lipid bilayers the excitation was done at even shorter wavelength (488 nm). This raises a question whether the authors really observe the same type of photobleuing in figures 1b and 1c,d. To clarify this, fluorescence spectra at 561 excitation should be also recorded before and after photobleuing (like it is done for Figure 1c,d). This could also help to identify better the emission maximum of the photoblued species. Moreover, the emission spectra for the studied lipid conjugates in supported bilayers before and after photobleaching should be also shown for both 640 and 488 nm excitation.

We realized that our notation on what kind of laser wavelengths were used was sometimes confusing, and we have now noted it deliberately. Following the reviewer's suggestion (to ensure we observe the same type of photobleuing in spectral and FCS experiments), we recorded the fluorescence spectra of lipid-conjugated dyes in supported lipid bilayers using 488 and 561 nm excitation in confocal and STED modes (new Extended Data Fig. 1e-f). Unfortunately, we could not perform FCS measurements upon 561 nm excitation, since the emission signal from the original red dyes was leaking into the blue-shifted detection channel giving rise to too much cross-talk (see Extended Data Fig. 1c, right). Similarly, we unfortunately could not perform the spectral recordings for 640 nm excitation (insufficient spectral filtering). Still, consistent with the previously observed small emission shift, our new spectral unmixing analysis showed that the emission maxima of photoblued species was 20-25 nm lower than that of the original dyes for both 488 nm and 561 nm excitation (new Extended Data Fig. 1g). Upon 488 nm excitation we found that a third small peak became apparent around 600 nm for ATTO 647N, which was not visible when exciting the sample with 561 nm light (new Extended Data Fig. 1f).

4) The use of term "sensor" usually stands for devices. Term "probe" would be more appropriate here.

We adapted the text accordingly.

Reviewer #3:

Remarks to the Author:

The paper by Dasgupta and co-authors investigates photoconversion induced artefacts in fluorescence microscopy. It is a short paper but it packs a lot in and makes a number of important points. With fluorescence microscopy becoming more and more turnkey and procedure based, it is timely to be reminded of the potential for imaging artefacts and the need to be aware to the possibility of these in any given imaging experiment.

The authors make a good selection of dye molecules for study: Atto 647N, Star red, and Atto 655, which are all commonly used dyes, although they are all red emitters and it would have been a nice inclusion to have another dye or two from a different part of the spectrum.

We thank the reviewer for their positive evaluation.

We agree on the importance of investigating a broader selection of dyes. However, in this work we decided to focus on dyes usually employed in STED microscopy, in the long-wavelength part of two-colour (STED-)FCS experiments, and in membrane spectral (STED) imaging, which are red-emitting dyes (excited at around 640 nm or 561 nm), and to observe how much photobleaching is biasing these experiments. Therefore, our selection covers some of the most used dyes in these applications (STAR RED, ATTO 647N, ATTO 655, and Nile Red-based NR12A and NR4A). Moreover, investigating photobleaching of green fluorophores would require blue detection (below 500 nm), not currently available in our STED-FCS, spectral and -FLIM microscopes. Nevertheless, we are currently planning on experiments to expand our studies and investigate photobleaching mechanisms in more detail, which will include experiments on the photoconversion of a larger number of dyes and fluorescent proteins. This however would be too much for the scope of this paper.

Even so, from the three dyes studied, some interesting results are obtained, especially around the phenomenon of photobleaching. Firstly, the photoconversion spectra of the dyes clearly show how emission from photobleached molecules has the potential to contribute to other detection channels, and in this context, it is good to know that the photobleaching products are dim. Secondly, it is interesting that Atto 655 does not photobleach. Can the authors comment on why it is resistant to this, and why the other two dyes are susceptible to photobleaching? And related to this, do the authors have any thoughts on why there is a greater effect on dyes conjugated to antibodies?

We explored the effect of conjugation by quantifying spectral and lifetime changes of lipid-conjugated dyes. While antibodies carry several conjugated dyes (which can induce additional photophysical processes such as quenching due to dye-dye interactions), lipids are conjugated to a single dye molecule. Spectral measurements and unmixing analysis show that dyes conjugated to lipids undergo photobleaching to a similar extent than dyes conjugated to antibodies (new Extended Data Fig. 1b, e-h). FLIM measurements on the other hand show a strong dependency on the conjugation strategy (new Extended Data Fig. 3f-g), as dyes conjugated to lipids do not show an increase in lifetime, suggesting that lifetime changes do not arise from photobleaching, but rather a different phenomenon, that we attribute to partial bleaching (new Extended Data Fig. 3d,e).

Additionally, our new experiments suggest that photobleaching involves photooxidative reactions, as photobleaching is drastically reduced in the absence of oxygen (new Extended Data Fig. 2i-j). Oxygen is known to increase photoreactivity when in singlet state, which is promoted by triplet-triplet interactions. For example, ATTO 655

undergoes very low intersystem crossing to the triplet state (Buschmann et al, 2009) (Extended Data Fig. 2d), and thus we speculate that the absence of triplet might result in generation of less reactive oxygen species, preventing photobleaching.

We now discuss these possible mechanisms in the manuscript:

“To explore possible mechanisms of the photobleaching pathway, we investigated dependencies on oxygen. Oxygen is known to have two major effects on the dyes’ photophysics: 1) quenching of the triplet state and therefore less photoreactions out of that state, and 2) increased general photoreactivity due to interaction with the dissolved oxygen, especially when in its singlet state^{2,10}. We consistently observed that, photobleaching was suppressed as a photoreaction in the absence of oxygen (Extended Data Fig. 2i,j), as highlighted before for other dyes^{6,9}. This on one hand indicates that the photobleaching pathway involve photooxidation (such as in photooxidative N-dealkylation processes¹⁰ or through singlet oxygen produced by quenching of the dyes’ triplet state¹⁰) and hardly evolves directly through the triplet state, and on the other hand that photobleaching is only one of many possible photoreaction pathways. Interestingly, our FCS data highlights a correlation between the extend of photobleaching and the amount of intersystem crossing and thus triplet yield, being lowest for ATTO 655 and largest for Abberior STAR RED (Extended Data Fig. 2d). Since singlet oxygen is generated through triplet-triplet energy transfer, we speculate that photobleaching is fostered via this mechanism.”

And

“These data were recorded on antibody-conjugated dyes. Here, a known phenomenon arises from high labelling degrees, i.e. of more than one dye conjugated to a single antibody, which results in quenching effects and thus fluorescence lifetime changes of the dyes conjugated in close proximity^{12,13}. Illumination will in this case result in an initial partial bleaching of one of the dyes and thus a rise in lifetime and an initial rise in fluorescence intensity followed by the drop due to photobleaching of all dyes, as observed in our case (Extended Data Fig. 3d,e). This fact and the observation that the lifetime change was only partially inhibited upon enzymatic oxygen depletion (Extended Data Fig. 3c), highlighted that lifetime changes are not a consequence of photobleaching but rather due to the outlined partial photobleaching. Consistently, we did not observe a lifetime increase upon illumination for lipids conjugated to single dyes, which instead showed a minor lifetime decrease (Extended Data Fig. 3f). Consequently, single-dye conjugation will serve as an artefact mitigation strategy in FLIM experiments.”

The points about FLIM and FRET and changes in lifetimes are well made. Often a small change in lifetime, such as that seen for Atto 647N, is interpreted as a meaningful result in terms of the experiment being undertaken but the observation that it could arise from photobleaching is important to know.

In the same vein, the observation of spectral shifts of a few nanometres are also often interpreted as a result and the authors clearly demonstrate that such shifts can arise from photobleaching. Restoring the dye to prove that the observed shifts are an artefact and not the result of packing changes in a membrane is an excellent demonstration of this point. Establishment of the Nile red derivative NR4A as a suitable dye for these applications is also a useful contribution.

Some example FCS curves and fits should be included in the extended data along with fitting results, especially the anomaly parameter values. Also, why was 5 μ s chosen for the triplet time for all dyes in FCS fitting? Aren't these values known or couldn't this component be allowed to be freely fit?

We now show sample FCS curves and their fitting for each dye (Extended Data Fig. 2a-c). We also show the anomaly value (Extended Data Fig. 2d), which is close to 1 and an indication of free-Brownian diffusion, as we now explain in the manuscript:

"Analysis of the data revealed no anomaly in diffusion (Extended Data Fig. 2d) and allowed us to probe Q and N for both species."

The triplet value of 5 us was obtained experimentally through independent FCS measurements, which we now clarify in the Methods section. The triplet time was subsequently fixed when fitting photobleaching data to minimize fitting uncertainty.

It would be useful to note in the text where a number of frames observation period is noted, what this is in terms of time, to save having to work this out from the experimental details. E.g. lines 41, 56, 66.

Following this suggestion, we adapted Extended Data Fig. 2e to also show the experiment time.

Decision Letter, first revision:

Dear Pablo,

Thank you for submitting your revised manuscript "Effects and avoidance of photoconversion-induced artefacts in confocal and STED microscopy" (N METH-BC53743A). It has now been seen by the original referees and their comments are below. The reviewers find that the paper has improved in revision, and therefore we'll be happy in principle to publish it in Nature Methods, pending minor revisions to comply with our editorial and formatting guidelines.

TRANSPARENT PEER REVIEW

Please note: we allow redactions to authors' rebuttal and reviewer comments in the interest of confidentiality. If you are concerned about the release of confidential data, please let us know specifically what information you would like to have removed. Please note that we cannot incorporate redactions for any other reasons. Reviewer names will be published in the peer review files if the reviewer signed the comments to authors, or if reviewers explicitly agree to release their name. For more information, please refer to our FAQ page.

ORCID

IMPORTANT: Non-corresponding authors do not have to link their ORCIDs but are encouraged to do

so. Please note that it will not be possible to add/modify ORCID IDs at proof. Thus, please let your co-authors know that if they wish to have their ORCID added to the paper they must follow the procedure described in the following link prior to acceptance:

Sincerely,
Rita

Rita Strack, Ph.D.
Senior Editor
Nature Methods

Reviewer #1 (Remarks to the Author):

The revised manuscript carefully and thoughtfully addressed my and other reviewers comments. I believe this paper will provide an important resource in efforts to consider these pathways during advanced microscopy efforts.

Reviewer #2 (Remarks to the Author):

In the revised manuscript, the authors addressed well all my concerns and made sufficient number of new experiments to support their claims. Now I can recommend this manuscript for publication in the present form.

Reviewer #3 (Remarks to the Author):

Comments raised by reviewers are addressed well. The results in deoxygenated conditions are a good addition and increase the paper's importance and relevance.

Final Decision Letter:

Dear Pablo,

I am pleased to inform you that your Brief Communication, "Effects and avoidance of photoconversion-induced artefacts in confocal and STED microscopy", has now been accepted for publication in Nature Methods. The received and accepted dates will be September 8, 2023 and April 30, 2024. This note is intended to let you know what to expect from us over the next month or so, and to let you know where to address any further questions.

Over the next few weeks, your paper will be copyedited to ensure that it conforms to Nature Methods style. Once your paper is typeset, you will receive an email with a link to choose the appropriate publishing options for your paper and our Author Services team will be in touch regarding any additional information that may be required.

Once proofs are generated, they will be sent to you electronically and you will be asked to send a corrected version within 48 hours. It is extremely important that you let us know now whether you will be difficult to contact over the next month. If this is the case, we ask that you send us the contact information (email, phone and fax) of someone who will be able to check the proofs and deal with any last-minute problems.

If, when you receive your proof, you cannot meet the deadline, please inform us at rjsproduction@springernature.com immediately.

Please note that *Nature Methods* is a Transformative Journal (TJ). Authors may publish their research with us through the traditional subscription access route or make their paper immediately open access through payment of an article-processing charge (APC). Authors will not be required to make a final decision about access to their article until it has been accepted. Find out more about Transformative Journals

Authors may need to take specific actions to achieve compliance with funder and institutional open access mandates. If your research is supported by a funder that requires immediate open access (e.g. according to Plan S principles) then you should select the gold OA route, and we will direct you to the compliant route where possible. For authors selecting the subscription publication route, the journal's standard licensing terms will need to be accepted, including self-archiving policies. Those licensing terms will supersede any other terms that the author or any third party may assert apply to any version of the manuscript.

If you are active on Twitter/X, please e-mail me your and your coauthors' handles so that we may tag you when the paper is published.

To assist our authors in disseminating their research to the broader community, our SharedIt initiative

provides you with a unique shareable link that will allow anyone (with or without a subscription) to read the published article. Recipients of the link with a subscription will also be able to download and print the PDF. As soon as your article is published, you will receive an automated email with your shareable link.

Please note that you and your coauthors may order reprints and single copies of the issue containing your article through Springer Nature Limited's reprint website, which is located at <http://www.nature.com/reprints/author-reprints.html>. If there are any questions about reprints please send an email to author-reprints@nature.com and someone will assist you.

Best regards,
Rita

Rita Strack, Ph.D.
Senior Editor
Nature Methods